# *Nematocida displodere* mechanosensitive ion channel of small conductance 2 assembles into a unique 6-channel super-structure *in vitro*

**Alexandra Berg** [1,2]*, **Ronnie P.-A. Berntsson** [2,3]*, **Jonas Barandun** [1]*

**1** Department of Molecular Biology, The Laboratory for Molecular Infection Medicine Sweden (MIMS), Umeå Centre for Microbial Research, Science for Life Laboratory, Umeå University, Umeå, Västerbotten, Sweden, **2** Department of Medical Biochemistry and Biophysics, Wallenberg Centre for Molecular Medicine, Umeå University, Umeå, Västerbotten, Sweden, **3** Wallenberg Centre for Molecular Medicine & Umeå Centre for Microbial Research, Umeå University, Umeå, Sweden

* alexandra.berg@umu.se (AB); ronnie.berntsson@umu.se (RP-AB); Jonas.barandun@umu.se (JB)

**Data Availability Statement:** The cryo-EM volume of the 6-channel super-structure has been deposited in the EM Data Bank with accession code EMD-19937.

## Abstract

Mechanosensitive ion channels play an essential role in reacting to environmental signals and sustaining cell integrity by facilitating ion flux across membranes. For obligate intracellular pathogens like microsporidia, adapting to changes in the host environment is crucial for survival and propagation. Despite representing a eukaryote of extreme genome reduction, microsporidia have expanded the gene family of mechanosensitive ion channels of small conductance (*mscS*) through repeated gene duplication and horizontal gene transfer. All microsporidian genomes characterized to date contain *mscS* genes of both eukaryotic and bacterial origin. Here, we investigated the cryo-electron microscopy structure of the bacterially derived mechanosensitive ion channel of small conductance 2 (MscS2) from *Nematocida displodere*, an intracellular pathogen of *Caenorhabditis elegans*. MscS2 is the most compact MscS-like channel known and assembles into a unique superstructure *in vitro* with six heptameric MscS2 channels. Individual MscS2 channels are oriented in a heterogeneous manner to one another, resembling an asymmetric, flexible six-way cross joint. Finally, we show that microsporidian MscS2 still forms a heptameric membrane channel, however the extreme compaction suggests a potential new function of this MscS-like protein.

## Introduction

Microsporidia compose a large group of highly adapted, obligate intracellular pathogens that modified their genomic repertoire for exceedingly efficient parasitism [1,2]. Their early divergence within the fungal kingdom and their fast evolutionary development [3] shaped a highly divergent phylum of unicellular eukaryotes [4]. The phylum of microsporidia infects a broad range of hosts, covering almost all animal taxa, including humans [5–7]. While generally

**Funding:** R.P-A.B: Swedish Research Council 2016-03599 & 2023-02423 https://www.vr.se/english.html Knut and Alice Wallenberg Foundation https://kaw.wallenberg.org/en J.B.: Swedish Research Council 2019-02011 https://www.vr.se/english.html European Research Council PolTube 948655 https://erc.europa.eu/homepage The funders did not play any role in the study.

**Competing interests:** The authors have declared that no competing interests exist.

ubiquitous, microsporidia thrive in agriculturally kept aquatic and terrestrial animals [6], compromising their health and leading to significant economic losses [8]. In humans, microsporidia can cause disease in otherwise healthy individuals but mainly provoke opportunistic infections in immunocompromised hosts, such as AIDS patients, organ transplant recipients, and cancer patients [6,9,10].

An infection cycle typically begins with the ingestion of dormant microsporidian spores by a host from contaminated sources [8,10]. The spores reach the gastrointestinal tract and germinate in response to an appropriate stimulus [11,12]. Spore germination triggers the explosive shoot-out of the invasion organelle, the polar tube, which injects the infectious cell content, the sporoplasm, into a host cell [13,14]. Intracellularly, the sporoplasm hijacks host nutrients [15–17] to differentiate into a multinucleated, proliferating cell, the meront, which further develops into a sporont. The sporont divides, synthesizes the spore coat, and develops into a sporoblast where organelle formation occurs. Finally, the sporoblast evolves into a mature spore which can start a new infection cycle [18].

The intricate and successful nature of microsporidian parasitism is a result of efficient genome organization and compaction, which makes them highly dependent on their host for replication and development [19–21] but, for example, nearly halves their energy requirements in the form of ATP [22]. To achieve this, microsporidia minimized or removed many enzymes involved in biosynthetic or metabolic pathways whose functions can be compensated through host system exploitation [2,16,19–21]. Proteins required to exploit the host but also to escape its immune system were gained through horizontal gene transfer (HGT) from bacteria [2] or the host [23,24] and modified via gene duplications [24].

One universally conserved protein family that has undergone an unusual evolution in microsporidia is the mechanosensitive ion channels of small conductance (MscS). MscS are membrane-embedded homo-heptameric proteins that exist in both eukaryotes and prokaryotes [2,25,26]. Their main function as "osmotic safety valves" is to prevent the cell from bursting during hypoosmotic shock [27]. MscS is best studied in *Escherichia coli* and can be structurally divided into two domains; the helical domain, containing the transmembrane domain, and the vestibule domain (**Fig 1A**). The membrane-bound part is formed by an N-terminal anchor and transmembrane helices (TMHs), TMH1, and TMH2, the latter of which constitute the membrane-tension sensor of the channel. The sensors are connected to the central pore, which further interacts with the vestibule, a cage-like structure with a narrow, crown-shaped end (**Fig 1A**) [28,29].

A reduction in extracellular osmolarity induces swelling of the bacterial cell, consequently stretching the membrane. The resulting membrane tension causes rearrangement of the anchor, the sensory domain, and channel-bound lipids, leading to expansion and opening of the MscS pore (**Fig 1**) [29,31]. This enables the efflux of water and selected ions to release turgor pressure and restore cellular homeostasis [27].

Microsporidian genomes harbor at least five genes encoding MscS-like proteins [2], which, according to phylogenetic analyses [2], were distinguished into two subfamilies designated MscS1 and MscS2: MscS1, derived from eukaryotes through lineage-specific expansion, and MscS2, originating from bacteria and likely acquired via HGT by the last common ancestor of microsporidia [2]. While the *mscS1* family is represented with $\geq 4$ genes in each microsporidian genome sequenced to date, only a single *mscS2* can be identified. Further, MscS1 resembles a canonical, eukaryotic MscS in size and domain architecture, whereas MscS2 is substantially truncated in primary sequence; a consequence of losing a part of the vestibule domain and at least two of the three TMHs [2,32], relative to the bacterial MscS. The extreme truncation at the N-terminus is, to our knowledge, unique to microsporidia. For example, protozoan MscS-like channels have at least two predicted TMHs, of which TMH2 and TMH3

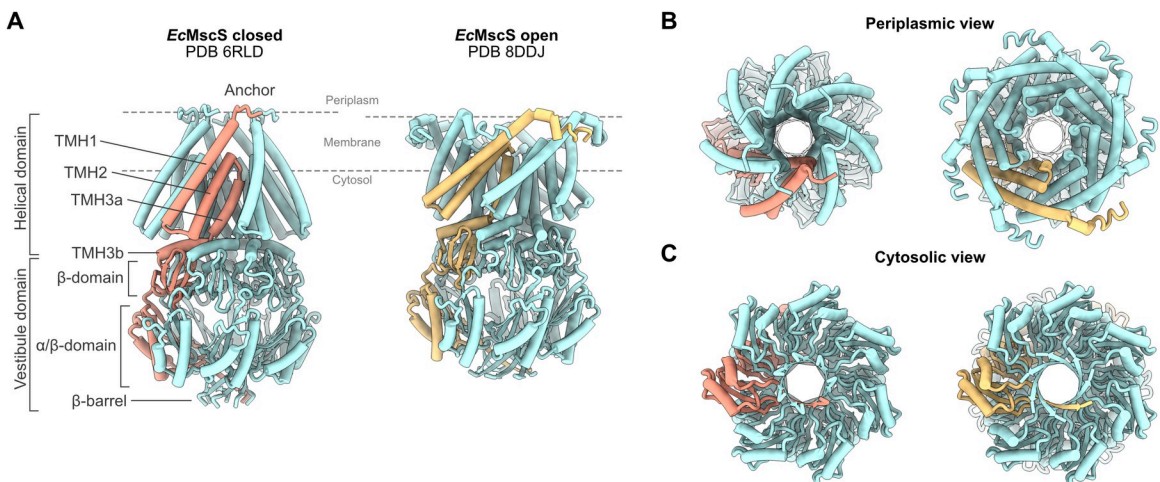

**Fig 1. Open and closed structure of *Escherichia coli* MscS. A)** Side view of MscS in a closed (PDB-ID 6RLD, left) and open (PDB-ID 8DDJ, right) conformation with one monomer indicated in tangerine (left) and pale yellow (right) to visualize structural rearrangements. Domains and secondary structures are labeled for the closed state. The inner membrane is indicated with dashed lines, according to Park et al. [30]. **B)** Periplasmic and **C)** cytoplasmic views of the closed and open conformations are displayed.

share high conservation with those of *Ec*MscS [33]. A smaller C-terminal domain is, however, also observed in *Arabidopsis thaliana* MSL-1, which lacks the cytosolic beta-barrel [34]. Thus, the extensive compaction phenomenon of the microsporidian MscS2 makes this evolutionary outlier an interesting target to study.

To understand the impact of compaction on the structure and function of microsporidian MscS2, we determined a low-resolution cryo-EM structure of MscS2 from *Nematocida displodere*, an intracellular pathogen of *Caenorhabditis elegans*. We show that MscS2 forms a homoheptameric channel that maintains two of the essential domains [35–37] required for channel gating. Under our *in vitro* conditions, the variant produces a higher-order superstructure, with six heptamers forming a structure that resembles an asymmetric, flexible six-way cross joint. While the observed superstructure likely is the result of applied experimental procedures and represents a physiologically non-relevant assembly, it serves as a unique and intriguing model that challenges our understanding of protein-complex formation. In addition, it shows that the minimal microsporidian MscS2 can still form a heptameric, membrane-bound channel with a central pore.

## Materials and methods

### Bioinformatics analyses

**Multiple sequence alignment.** Microsporidian MscS2 were identified through blasting against the non-redundant NCBI database, against one at the time unpublished microsporidian genome of *Vairimorpha necatrix* [38], and against a local database created from three available but non-annotated genomes of *Antonospora locustae* [39–41]. Two multiple sequence alignments of 35 identified sequences with and without *E. coli* MscS were generated in ClustalW (accessed September 2023) [42] with default settings and colored using Clustal X default Colouring in Jalview (v2.11.2.6) [43].

**Cladogram generation.** 35 MscS2 sequences were aligned using MUSCLE (v5.1) [42] and trimmed with trimAl (v1.4.1) [44] (to remove spurious sequences and poorly aligned regions from a multiple sequence alignment). The resulting sequences were used to generate a

cladogram with IQ-TREE (v2.0.3) [45,46] with 1000 bootstrap replicates and the MFP option for choice of substitution model.

Sequences used for the multiple sequence alignment and cladogram: *Annicaliia algerae*, *Antonospora locustae*, *Dictyocoela muelleri*, *Edhazardia aedis*, *Encephalitozoon cuniculi*, *Encephalitozoon hellem*, *Encephalitozoon intestinalis*, *Encephalitozoon romaleae*, *Enterocytozoon bieneusi*, *Enterocytozoon hepatopenaei*, *Enteropsecta breve*, *Enterospora canceri*, *Hamiltosporidium magnivora*, *Hamiltosporidium tvaerminnensis*, *Hepatospora eriocheir*, *Nematocida ausubeli*, *Nematocida displodere*, *Nosema granulosis*, *Nematocida homosporus*, *Nematocida major*, *Nematocida parisii*, *Nematocida species ERTm2*, *Nematocida species ERTm5*, *Ordospora colligata*, *Ordospora pajunii*, *Pancytospora epiphaga*, *Pancytospora philotis*, *Spraguea lophii*, *Thelohania contejeani*, *Trachipleistophora hominis*, *Tubulinosema ratisbonensis*, *Vairimorpha ceranae*, *Vairimorpha ceranae BRL01*, *Vairimorpha necatrix*, *Vavraia culicis*.

**Topology prediction.** For transmembrane region and signal peptide predictions of the 35 MscS2, we used DeepTMHMM (v1.0.24) [47] and TOPCONS (v2.0) [48]. All sequences were entered individually to obtain a result graph for each protein.

**Hydropathicity prediction.** To identify primary-sequence based hydrophobic regions in *N. displodere* MscS2, we employed ProtScale on the ExPASy server [49] to predict the hydropathicity [50] of the protein.

**Alphafold prediction.** The AlphaFold models for *N. displodere MscS2* were generated with ColabFold (v1.5.5) [51]. First, a single-chain model was predicted using the default model type alphafold2_ptm. The heptamer model was created using the first-rank monomer as a template, with the alphafold2_multimer_v3 model type. For both predictions, we employed 20 prediction cycles and an early stop tolerance of 0.2. To display heptamer model confidence, we colored the model according to the pLDDT (predicted local-distance difference test) confidence measure [52].

**Cloning, protein production and bacterial cell lysis.** We ordered *E. coli* codon-optimized *N. displodere mscS2* from Genewiz (Azenta Life Sciences) in a pUC-GW-Kan vector. Using primers listed in S1 Table, we amplified *N. displodere mscS2*, *mscS2Δ2–11* and *mscS2Δ2–29* via polymerase-chain reaction and cloned each gene into a *pRSFDuet* vector with an N-terminal His14-SUMO tag and a kanamycin resistance cassette. We transformed chemically competent *E. coli TOP10* cells with each plasmid, respectively, for propagation, validated positive clones using colony-PCR and sequencing. Subsequently, we transformed *E. coli* Rosetta (DE3) cells with the vector. For protein production, we inoculated 100 mL Luria Bertani (LB) medium containing 50 µg/mL kanamycin and 33 µg/mL chloramphenicol with colonies carrying the respective plasmid and grew them overnight at 37˚C with shaking. Next, we inoculated the main culture 1:35 in LB and further incubated the culture at 37˚C and 150 rpm. When reaching OD600nm 0.7, the temperature was shifted to 18˚C and induced expression by the addition of 1 mM IPTG for 18 h. The cells were then pelleted and frozen in liquid nitrogen before being lysed using a cryo-mill (Retsch), with three cycles of milling. The resulting bacterial powder was stored at -80˚C.

**Protein purification.** To purify MscS2Δ2–29, we thawed 5 g of the bacterial powder and resuspended it in 20 mL buffer A [50 mM HEPES-KOH, 300 mM NaCl, 10% glycerol, pH 8 at 4˚C], added 40 mM imidazole, 1 mM DTT, 1 mM MgCl$_2$, DNase and protease inhibitors Phenylmethanesulfonyl Fluoride [10–20 µM], Pepstatin [1–2 µM] and E64 [1–2 µM]. We used a tissue grinder (Dounce homogenizer) to homogenize the suspension before adding 1.5% (w/v) n-Dodecyl-beta-Maltoside (DDM) and incubating 1 h at 4˚C for solubilization. The insoluble fraction was removed by spinning down the sample at 48,500 x g for 1 h at 4˚C. The supernatant was loaded on a His-Trap 5 mL Ni-NTA column (Amersham Biosciences), washed three times with 10 column volumes buffer A containing 60 mM, 80 mM, and 100 mM imidazole

and 0.05% (w/v) DDM, respectively. The protein was eluted by buffer A containing 500 mM imidazole and 0.05% (w/v) DDM. Purification fractions were analyzed using sodium dodecyl sulfate polyacrylamide gel electrophoresis (SDS-PAGE). Next, to cleave the His14-SUMO tag we pooled all elution fractions that contained His14-SUMO-MscS2Δ2–29, transferred it to a 7 kilodalton cut-off SnakeSkin™ dialysis tubing (Pierce™, VWR), added Ulp1 protease [0.5 μM final concentration] and let it dialyze overnight at 4°C in buffer A (with 7% glycerol) and with stirring. We assessed the successful cleavage via SDS-PAGE and split the sample in two. We analyzed one half using size-exclusion chromatography and dialyzed the other half again in buffer A without glycerol and with 0.05% DDM, to prepare MscS2Δ2–29 for cryo-EM grid freezing. For gel filtration experiments of MscS2Δ2–29 we used a Superdex 200 Increase 10/300 GL (Cytiva) set up on an ÄKTA™ pure (Cytiva). We equilibrated the column with buffer A containing 0.05% DDM and 1 mM DTT without glycerol and loaded 500 μL pooled and cleaved elution sample.

**Negative-stain transmission electron microscopy.**   MscS2Δ2–29 was stained with 1.5% uranyl-acetate (EMS), as follows: We glow-discharged copper grids (200 squared mesh) coated with a thin carbon film for 30 sec at 15 mA, applied 3 μL of sample (concentration of 0.194 mg/mL), let it air dry for 30 sec, blotted off excess sample with Whatman paper, washed three times in 20 μL- drops of MiliQ water, stained for 30 sec in a 20μL-drop of uranyl acetate and remove excess stain with Whatman paper. Prior to storage at room temperature, we let the sample air dry for ~15 minutes. We analyzed the MscS2Δ2–29 grids using a FEI Talos L 120C transmission electron microscope (Thermo Fisher Scientific), operating at 120 kV, and acquired TEM micrographs at a magnification of 73,000 with a Ceta 16M CCD camera employing the TEM Image & Analysis software (v4.17) (FEI).

**Cryo-EM grid preparation and data collection.**   For cryo-EM analysis of MscS2Δ2–29, we applied 4 μL aliquots (0.33 mg/mL) to glow-discharged (30 sec, 15 mA) Quantifoil R2/2 200-mesh copper grids with 2 nm carbon (EM sciences) at 4°C and 100% humidity using the FEI Vitrobot Mark IV (Thermo Fisher Scientific). The sample was blotted for 5s, with -5 blot force prior to vitrification in liquid ethane.

Cryo-EM data were collected at the Umeå Core Facility for Electron Microscopy employing a 200 kV Glacios system (Thermo Fisher Scientific) equipped with a Falcon 4i direct electron detector. For MscS2Δ2–29, a total of 2000 movies, each with 40 frames, at a total dose of 50 e-/Å2, and a pixel size of 0.75 Å were collected using EPU (Thermo Fisher Scientific). We summarized the data collection statistics in **S2 Table**.

**Cryo-EM data processing.**   We processed the MscS2Δ2–29 cryo-EM data with Relion version 3.1 [53], 4.0 [54] and 5.0 beta [55]. Movie alignment, drift correction, and dose weighing were performed using MotionCor2 [56], and CTF estimation was done with CTFFIND (v4.1.14) [57]. We manually inspected and removed micrographs with poor CTF or low ice quality resulting in 1856 micrographs from which 464,200 particles were extracted with a box size of 300 px using Laplacian autopicking in Relion. Particles were subjected to 2D classification, and well-defined class averages were used for reference-based picking on the 1856 micrographs. The 737,506 picked particles were extracted with a box size of 384 pixels, and suboptimal classes were filtered out through consecutive 2D classification. From the resulting 351,941 particles, we generated an initial model and performed a 3D classification to find the best 3D class. The 157,712 particles contributing to this class were subjected to a consensus refinement without imposed symmetry leading to a 6-channel superstructure of 8.3 Å resolution. To receive a better volume of a single channel, we performed a focused 3D refinement on the two best-resolved channels, which resulted in a 7.7 Å volume. A processing scheme is illustrated in S1 Fig.

Other techniques that we applied to obtain a higher-resolution volume or sub-volume comprised non-uniform and heterogeneous refinement (CryoSPARC [58]), masked 3D classification of one, two, and three channels, focused refinement of a single channel, consensus refinement with D3 symmetry, symmetry expansion (RELION 5.0 beta), and 3D variability analysis (CryoSPARC). Representative approaches are illustrated in **S2 Fig**. The overall issue with this dataset is the heterogeneity of the complex, which results in poor particle alignment. Therefore, none of the applied methods led to an improved cryo-EM volume compared to the one obtained through data processing, documented above and in **S1 Fig**.

**Mass photometry.** For mass analysis, we used a RefeynMP2 instrument. We first cleaned a microscope cover slip (No. 1.5, 24x50 mm, Thorlabs) through alternated rinsing with MiliQ $H_2O$ and isopropanol, finishing off with MiliQ $H_2O$ and drying them under a clean nitrogen stream. On the center cover slip, we applied a silicone gasket, a self-adhesive frame with culture wells, to hold the later droplet sample. We diluted MscS2Δ2–29 samples in buffer [50 mM HEPES-KOH, 300 mM NaCl, pH 8 at 4°C] to concentrations of 28 nM and 56 nM immediately before measurements. Buffer only was used as control. Each sample was applied to a fresh well, the focal position was determined, and movies of 60 sec duration were recorded. For data acquisition and evaluation, we used Refeyn Acquire MP and DiscoverMP software, and NativeMarkTM (ThermoFisher) was used as standard.

## Results and discussion

### Microsporidia retain TMH3 and the beta domain in MscS2

We used different bioinformatics tools to identify MscS2 across microsporidia, obtain sequence information, and predict domains and features to analyze the impact of compaction on structure and potential function. We identified 35 MscS2 through multiple rounds of blasting against the non-redundant NCBI database, one at-the-time unpublished microsporidian genome from *Vairimorpha necatrix* [38], and a local database created from three available but non-annotated genomes of *Antonospora locustae* [39–41].

A multiple sequence alignment of these 35 MscS2 confirmed earlier findings that the MscS β-domain is conserved among microsporidia (**Fig 2A**). Residues located in important structural positions in the β-domain, e.g., the kink-inducing P129, G143, P166, are conserved between microsporidia and *E. coli* (**S3A Fig**), suggesting a similar tertiary fold. Further, the region just before the MscS β-domain, which in *E. coli* MscS (*Ec*MscS) corresponds to the amphipathic transmembrane helix (TMH) 3b, also shows reasonably high conservation (**Figs 2A and S3A**). To identify potential TMHs and signal peptides (SPs) in MscS2, we used TOPCONS and DeepTMHMM. However, the results were inconclusive, as the tools suggest the presence of one or more TMHs for only 13 MscS2 proteins while predicting no TMH for eight MscS2 and contradicting each other on the remaining sequences (**Fig 2B**). Furthermore, the tools predict an SP for 25–50% of the sequences. However, TMHs are often mistaken for SPs in these kinds of analyses [59,60]. Additionally, for some MscS2, the sequence length just before the MscS β-domain seems insufficient to accommodate two TMHs, which are typically at least 17 aa long [61,62]. For example, the sequence length of Enterocytozoonidae and close neighbors, as well as *Trachipleistophora hominis* MscS2, range from 73–95 aa (**Fig 2B**), leaving only 22–45 aa for one or two potential TMHs and a connecting loop. Taken together, based on primary sequence, it remains elusive how many TMHs microsporidian MscS2 have.

Moving from primary sequence to 3D protein structure, we created an AlphaFold model of *N. displodere* MscS2. While the model is of high confidence in the MscS β-domain and the adjacent helix, the protein fold and domain orientation of the remaining N-terminus, however, are of low confidence (**S3B Fig**). A hydropathicity prediction of the N-terminal MscS2

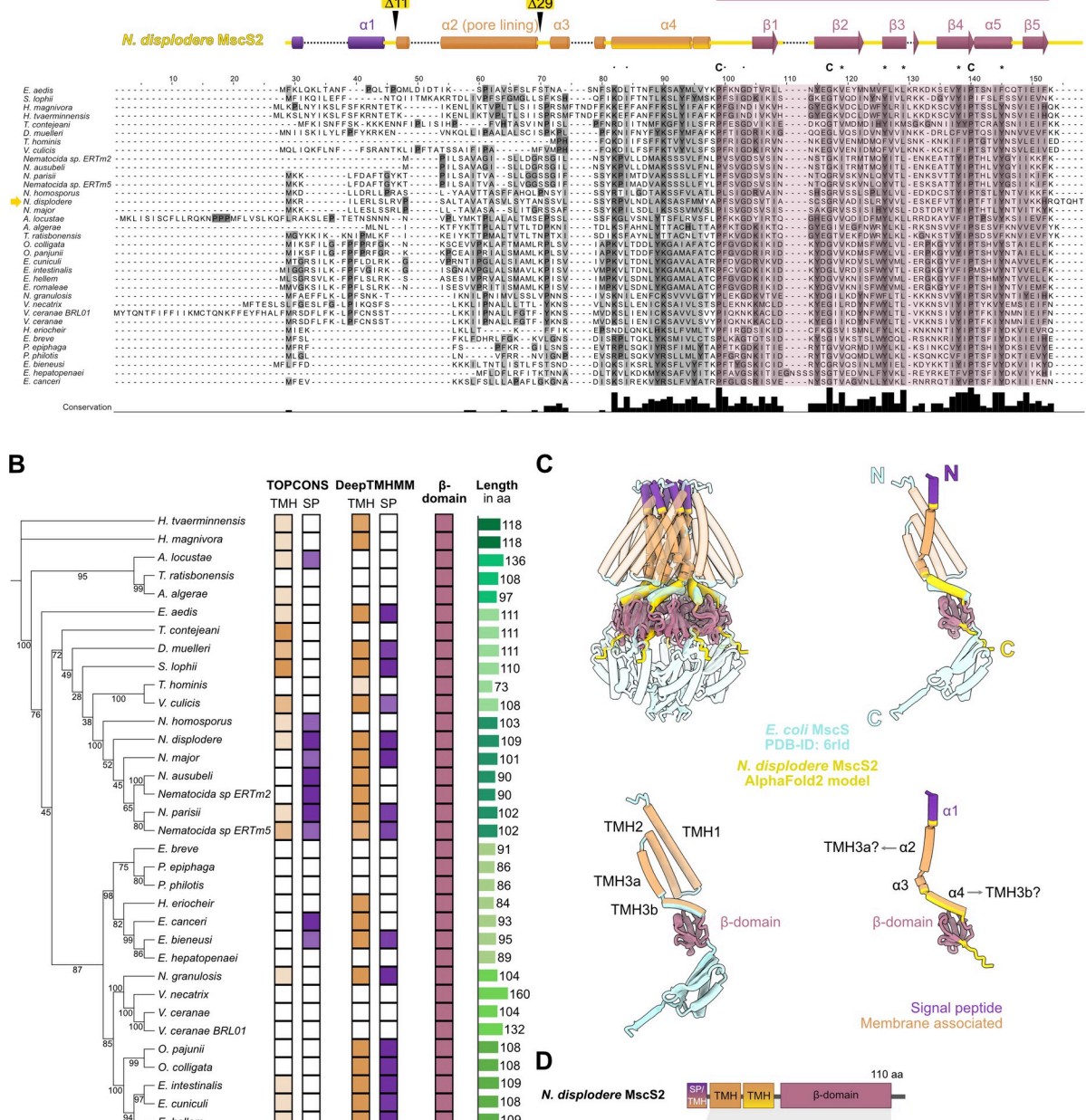

**Fig 2. MscS2 is highly conserved among microsporidia and drastically shortened to the minimally required ion-channel domains. A)** Sequence alignment of MscS2 from all sequenced microsporidian species and indicated secondary structure elements of *Nematocida displodere* MscS2 with truncation marks. The conserved MscS b-domain is highlighted in shades of pinkish red. Residues marked with "C" are fully conserved, residues with "*" are conserved between groups with very similar characteristics, and residues with "." are conserved between groups with weakly similar characteristics. Multiple sequence alignment was done using ClustalW with default settings and colored according to conservation with "Clustal" in Jalview (v2.11.2.6) and set to black and white. a, alpha-helix; b, beta-strand; MscS, mechanosensitive ion channel. **B)** Cladogram of microsporidia MscS2 indicating the presence of one or more transmembrane helices (TMHs), a signal peptide (SP), the MscS b-domain, and the amino acid sequence length. **C)** Structural comparison and superimposition of *N. displodere* MscS2 Alphafold2 model with *E. coli* MscS (PDB-ID 6RLD) channel and single chain. N, N-terminus; C, Carboxy-terminus; TMH, transmembrane helix. All membrane interacting structures are indicated in orange, all signal peptides are colored violet, and all MscS b-domains are marked pinkish red. **D)** Schematic domain architecture of *N. displodere* MscS2, representative of microsporidia, and *E. coli* MscS, representing bacteria. The grey area indicates the conserved region.

sequence (**S3C Fig**) indicates the presence of hydrophobic residues suggesting a transmembrane region. To assess structural similarities and differences between MscS2 and a bacterial homolog, we compared the AlphaFold model to the structure of *Ec*MscS. A superimposition of both proteins highlights the conserved MscS β-domain fold and indicates that the MscS2 helices might correspond to TMH3a (pore-lining) and TMH3b in the bacterial homolog (**Fig 2C**). In *Ec*MscS, TMH3 was stated to be essential for channel gating function [36,63]. To summarize, microsporidian MscS2 is heavily compacted and lacks N-terminal TMH1 and possibly TMH2, which, based on shorter size, might correspond to a signal sequence. However, with TMH3 and the conserved β-domain, the microsporidian MscS2 maintains two essential domains of an ion channel (**Fig 2D**).

### *Nematocida displodere* MscS2 forms a 400-kDa assembly

Next, we used the obtained information on structure prediction and architecture to design constructs for recombinant MscS2 production. To produce MscS2 and analyze it via single-particle cryo-EM, we ordered codon-optimized *mscS2* from four different microsporidian species: *Nematocida displodere*, *Encephalitozoon cuniculi*, *A. locustae*, and *Enterocytozoon bieneusi*. Out of those, *mscS2* from *N. displodere* produced the highest amount of protein (**S4A Fig**) and was used for large-scale protein production and purification. Based on our bioinformatics analyses, *N. displodere* MscS2 (hereafter referred to as "MscS2" or "*mscS2*") is a transmembrane protein, with one or two transmembrane helices, depending on whether the first is a signal peptide (SP) or not. We thus designed three different *mscS2* constructs, codon-optimized for production in *E. coli* cells: A full-length version of *mscS2* and two N-terminally truncated versions, *mscS2Δ2–11* and *mscS2Δ2–29*, which lack part of, or the whole predicted SP sequence, respectively.

We performed initial solubilization trials of MscS2 variants indicating that DDM and LMNG are the most effective detergents for solubilization. The purifications of pure MscS2 and MscS2Δ2–11 were unsuccessful due to degradation and significant aggregation after solubilization of both variants. In addition, the amounts of these two variants were substantially lower than that of MscS2Δ2–29, which could readily be purified to homogeneity in both DDM and LMNG. Initial size-exclusion chromatography (SEC) analysis of the purified His14-SUMO-MscS2Δ2–29 shows a main peak corresponding to a predicted molecular mass of around 320 kDa (according to elution volume), both when solubilized with DDM and LMNG (**S5A Fig**). After tag cleavage, the SEC profile shows a single symmetric peak which corresponds to an even higher molecular weight of circa 430 kDa (**Figs 3A and S5A**). This indicates that the His14-SUMO tag poses a steric hindrance at the N-terminus, thereby preventing higher oligomer formation. The predicted mass from SEC is > 6-fold higher than the theoretical weight of 64 kDa for the expected homo-heptamer (lacking the weight from the detergent micelle). Visualization of the SEC peak fractions via SDS-PAGE indicates a single protein band corresponding to MscS2Δ2–29 monomers (**Figs 3A and S5B**), suggesting that MscS2Δ2–29 oligomerize and form a defined high molecular weight protein-detergent complex. To assess whether this assembly is concentration-dependent, we diluted the protein sample and analyzed it using mass photometry. The mass photometry analysis showed a main peak corresponding to 400 kDa (28 nM) and 375 kDa (56 nM) (**Fig 3B**). The peaks of low molecular weight correspond to background noise (**S5F Fig**). This indicates that the assembly remains stable even when diluted and confirms the unexpectedly high mass indicated by the SEC.

To gain more insight into the large oligomeric assembly, we performed negative-stain transmission electron microscopy (TEM) to assess particle abundance and purity.

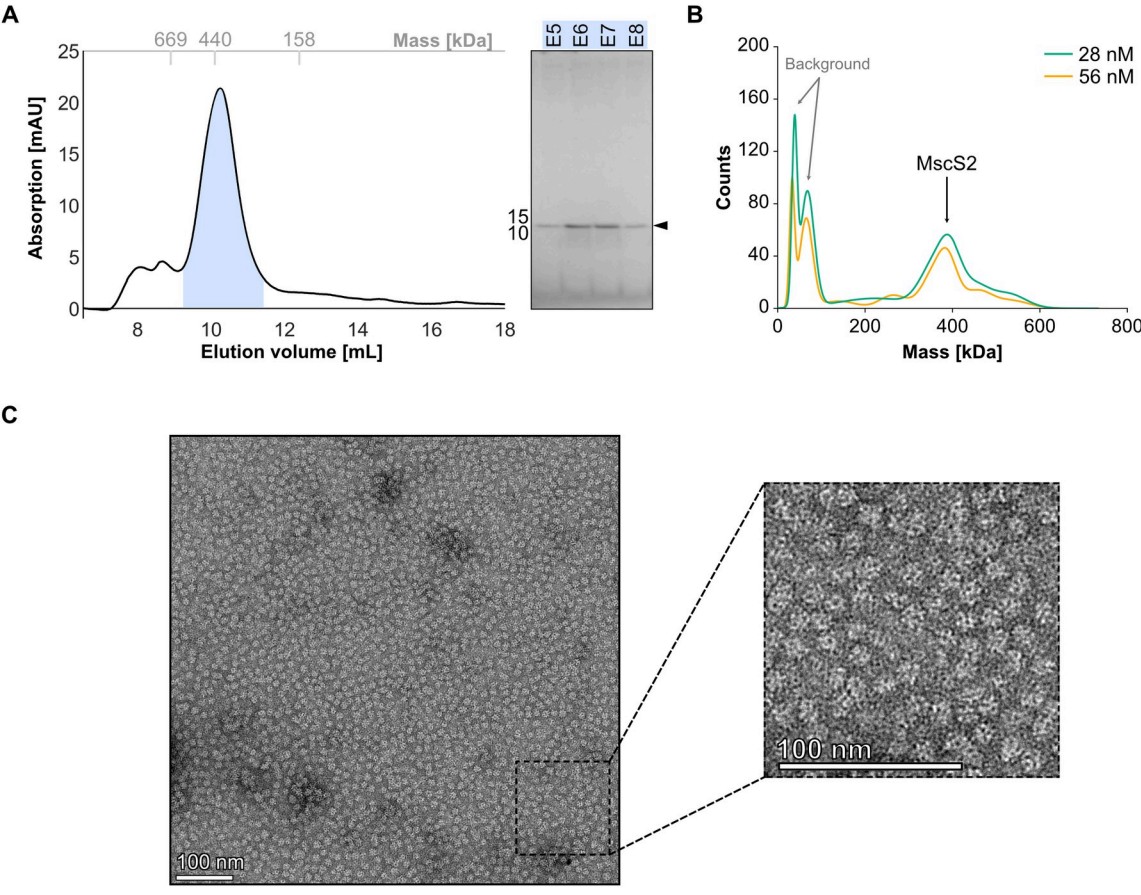

**Fig 3.** *Nematocida displodere* **MscS2Δ2–29 assemble into a high-molecular-weight complex. A)** Size exclusion chromatography (SEC) elution profile of MscS2Δ2–29 [15μM] sample on a Superdex 200 increase 10/300 GL column with SDS-PAGE analysis of the main peak fractions highlighted by a blue shade. The x-axis of the SEC profile shows the elution volume in milliliters (mL), the y-axis displays the absorption in milli-absorbance units, and the top x-axis (grey) indicates the molecular weight of standard globular proteins. **B)** Mass photometry analysis of MscS2Δ2–29 indicating a molecular weight of 400 (±60) kDa and 375 (±30) kDa for MscS2Δ2–29 at concentrations 28 nM and 56 nM, respectively. **C)** Representative transmission electron microscopy (TEM) micrograph of negative-stained MscS2Δ2–29 particles with zoom into the lower right corner. Scale bar diameter is 100 nm.

Confirming the high mass observed in SEC and mass photometry, the particles were significantly larger and different in shape (**Fig 3C**) compared to MscS particles from *E. coli* visualized via TEM [31,64], suggesting an unusual oligomeric state.

## MscS2 forms a homo-heptameric channel that assembles into a six-channel super-structure

While the negative stain micrographs do not show the expected heptameric pore-shaped particles, they confirm the presence of a homogeneous, large protein complex, as seen in the SEC and mass photometry analysis. To further characterize this high molecular weight complex formed by MscS2Δ2–29, we collected a single-particle data set using cryo-EM. We observed similarly shaped particles (**Fig 4A**) as before with negative stain EM (**Fig 3C**). A 2D classification of 351,941 particles resulted in unusual, wind-mill-shaped 2D class averages (**Fig 4B**). Further 3D classification and a consensus 3D refinement of the best class without imposing symmetry led to an 8.3 Å volume. The determined volume resembles an asymmetric, flexible six-way cross-joint (**Fig 4C–4E**). There, six individual channels are oriented in a

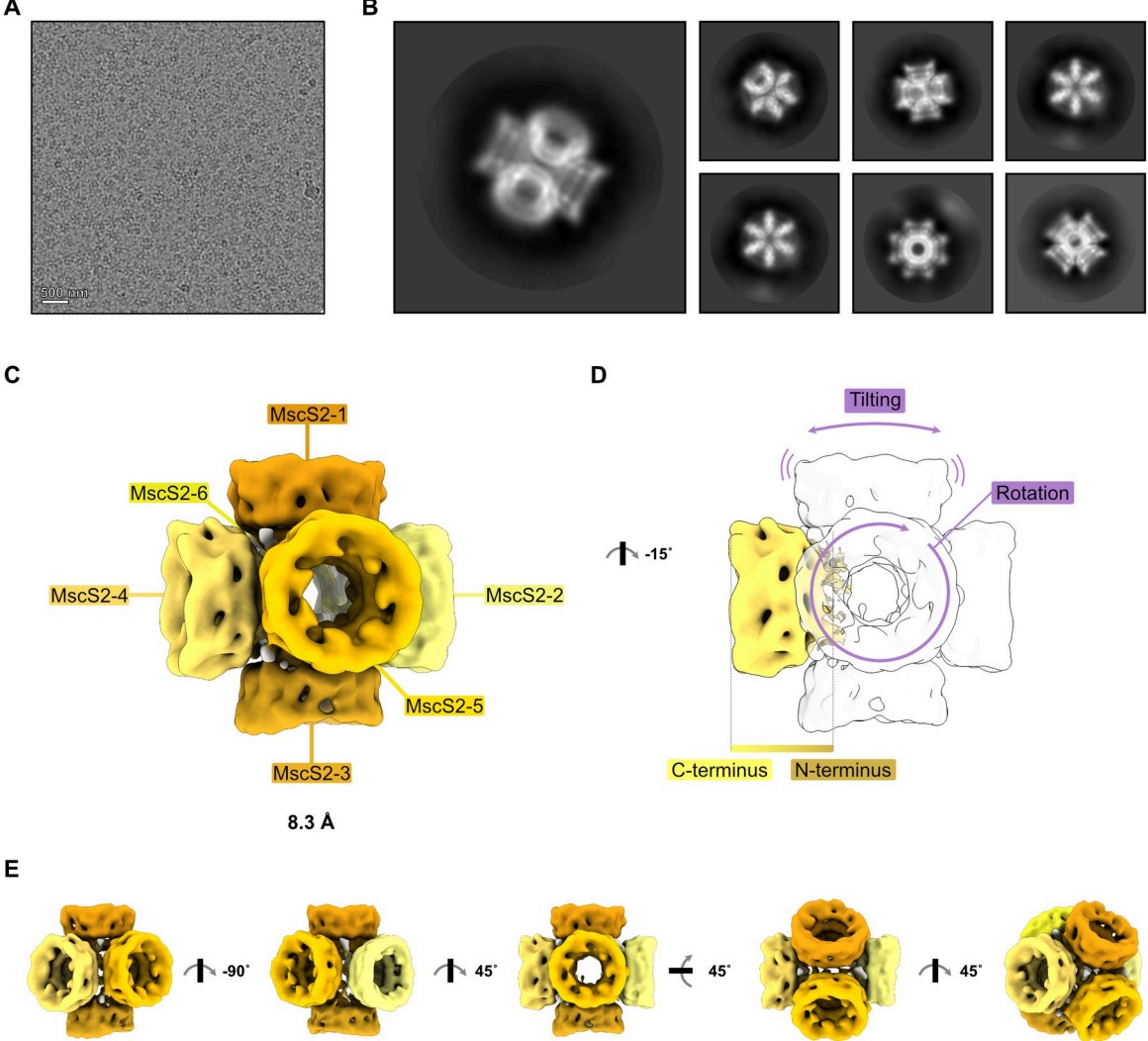

**Fig 4.** *Nematocida displodere* **MscS2Δ2–29 assembles into a six-channel superstructure** *in vitro*. **A)** Representative cryo-EM micrograph of the MscS2Δ2–29 sample. **B)** Representative 2D class averages of MscS2Δ2–29 particles. **C)** MscS2Δ2–29 cryo-EM volume colored in shades of yellow with labeled heptamer channels and resolution in Å. **D)** Rotationally related view of panel C) with one channel colored to visualize the directionality, N-terminus oriented towards the center and the C-terminus outwards. Heterogeneity and flexibility of individual MscS2Δ2–29 channels are indicated with purple arrows. **E)** Rotationally related views of the MscS2Δ2–29 superstructure.

heterogeneous manner to one another with the N-termini facing inwards (**Fig 4D**) and seem to oligomerize via their truncated transmembrane domains. We were unable to achieve a higher resolution of the superstructure due to inherent flexibility within the protein complex. Presumably, each of the six channels per complex is rotated and/or tiled at a different angle, which makes every particle unique and hampers particle alignment (**Fig 4D**).

The tight packing of the channels suggests that the presence of the first 29 amino acids, which were truncated to remove the predicted SP and to obtain a homogeneous protein sample, would not allow MscS2 to form this unusual assembly. However, we assume that the only stable condition of the truncated MscS2 channels *in vitro* is this 6-channel superstructure, as electron microscopy and mass-determination experiments are dominated by the large oligomeric complex.

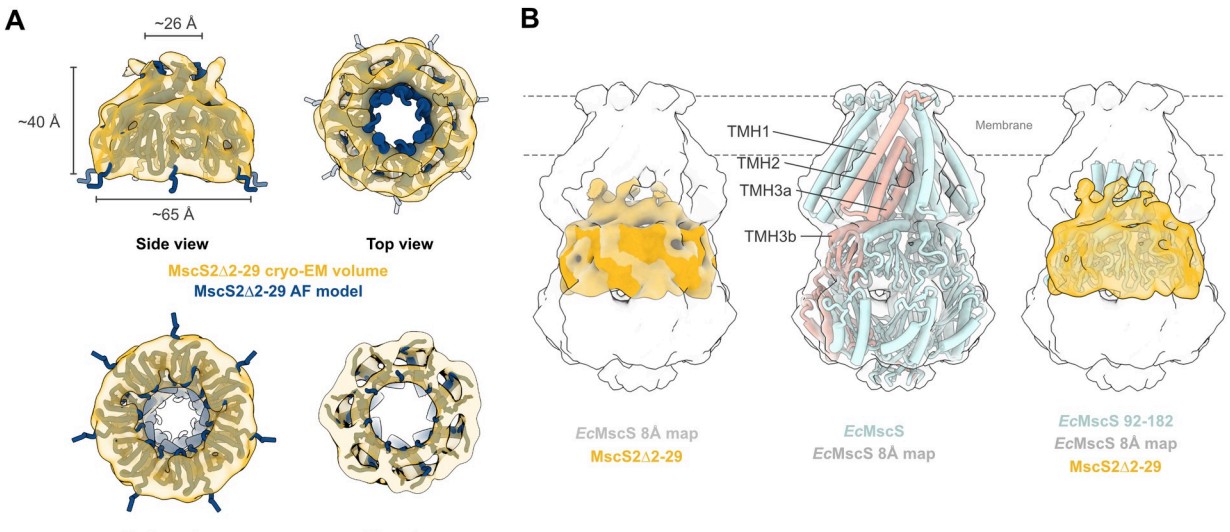

**Fig 5.** *Nematocida displodere* **MscS2 cryo-EM volume embeds well in the *Ec*MscS β-region. A)** Top panels: Side and top view of *N. displodere* MscS2Δ2–29 volume (goldenrod transparent) superimposed with the corresponding AlphaFold model (dark cerulean). Approximate inner and outer diameter and height of the volume (σ-level 0.006) are indicated. Bottom panel: Bottom and slice view of the cryo-EM volume superimposed with the AlphaFold model. **B)** *Ec*MscS (6RLD) low-pass filtered map (light gray transparent) superimposed with the MscS2Δ2–29 volume (left panel), *Ec*MscS structure in the 8 Å map (middle) with one monomer colored in tangerine and labeled TMHs, and a truncated version of *Ec*MscS superimposed with the MscS2Δ2–29 channel volume within the *Ec*MscS 8 Å map (right).

Next, we extracted one subunit of the 6-channel superstructure to assess its individual oligomeric state. Due to the SP truncation, the single-channel volume ds mostly for the cytoplasmic domain, including the TMH3b and beta-domain, and only a small portion of the potential pore-lining helix, TMH3a. We superimposed the single-channel volume with the predicted AlphaFold model of the heptameric, full-length MscS2 (with the first 29-aa hidden) (**Fig 5A**). The AlphaFold homo-heptamer fits into the obtained cryo-EM volume, supporting the hypothesis that *N. displodere* MscS2 indeed forms a homo-heptamer (**Fig 5A upper right** and **lower right panel**), like the bacterial homolog [65]. The MscS2Δ2–29 volume has a pore size of ~26 Å, measured within the MscS β-domain, an outer diameter of ca. 65 Å, and is about 40 Å in length along its central axis (**Fig 5A upper left panel**). To compare it to the bacterial homolog, we superimpose it with an 8 Å low-pass filtered map of *Ec*MscS (PDB id: 6RLD) (**Fig 5B left panel**). The compaction of MscS2 becomes very evident as it only fills out one-third of the filtered *Ec*MscS map, even considering that one potential TMH was removed. However, the MscS2Δ2–29 volume overlays nicely with the "middle section" of *Ec*MscS, the MscS β-domain, and the amphipathic TMH3b (**Fig 5B right panel**). In *Ec*MscS, TMH3b has been shown to interact with the β-domain and is essential for channel gating function [36]. Therefore, it is reasonable to assume that MscS2 has the prerequisites for a functional yet likely different gating mechanism. How the full-length MscS2 embeds in the membrane bilayer to sense membrane tension, if it still has the capacity to do so, could not be determined in this study. In *Ec*MscS, TMH1 and TMH2 form the tension sensor paddle of which the top part embeds into the membrane (**Fig 5B middle panel**) [28,29,66,67]. In contrast, the predicted transmembrane domain of full-length *N. displodere* MscS2 only has enough residues to pass through the membrane once (**Fig 2A**). Thus, it is unclear whether this compacted MscS-like protein would function like the bacterial homolog. To assess this, one would need to obtain full-length MscS2 protein and perform electrophysiological experiments to analyze potential mechano-sensation characteristics.

Another difference between the bacterial and microsporidian channel is the partially missing vestibule domain in MscS2 [2,68]. In *Ec*MscS, it harbors the α/β-domain and the β-barrel, which confer ion selectivity [67,68]. *Ec*MscS variants lacking the β-barrel remained active under hypoosmotic shock but were less abundant or less stable in the membranes compared to wild-type MscS, while *Ec*MscS variants missing both domains were not even inserted into *E. coli* membranes [37]. It is, therefore, puzzling how microsporidia would employ such a reduced version of MscS2. One possibility is that MscS2 in microsporidia does not have any ion selectivity.

Considering the extreme compaction of MscS2, the lack of the archetypical MscS sensing paddle, and the much smaller C-terminal domain, it is possible that microsporidia transformed MscS2 into some other type of channel beneficial for their intracellular life cycle. This hypothesis could be supported by the presence of at least four genes encoding canonical MscS1 in microsporidian genomes, although their function as mechanosensitive channels has not yet been studied [2].

*mscS2* is, as previously mentioned, present in all microsporidia, and transcriptional data shows that it is expressed and, therefore, likely functional (**S6 Fig**). Transcriptome analyses of *Edhazardia aedis*, *Vavraia culicis* [69], *Encephalitozoon cuniculi* [70] and *Nematocida parisii* [1] suggest that *mscS2* is predominantly upregulated during proliferative stages, such as the schizogony and merogony, but also moderately expressed during stages that require morphological changes, like sporogony, sporulation and nuclear dissociation (**S6 Fig**). However, further experiments are necessary to elucidate the detailed structure and function of the protein. Suggestions for future research include raising antibodies against MscS2, which could aid in locating the channel in infected *C. elegans* as well as pinpointing at which stage of the microsporidian lifecycle MscS2 is produced. Additionally, such antibodies could stabilize the channel and aid in purification, and especially structure determination. Further, it will be exciting to uncover if, and in that case, how microsporidia can operate this minimal version of MscS2 with regards to the sensing and gating mechanism.

Our findings show that *N. displodere* MscS2 forms a homo-heptameric channel that retains the MscS β-domain and a TMH3-like helix. We also show a remarkable complex assembly, which, despite not being physiologically relevant, provides fascinating information on the MscS2 assembly as well as on how membrane proteins can oligomerize when not restricted to the lipid membrane.

## Supporting information

**S1 Fig. Cryo-EM data collection and processing.** Initially, 2000 micrographs were collected and manually inspected to remove any affected by drift, poor CTF fits, or low-quality ice, resulting in a total of 1856 micrographs. Particles were picked through Laplacian auto-picking, followed by 2D classification and 2D class average-based template picking on all 1856 micrographs. Consecutive 2D classification was used to eliminate picking contaminants and sort out suboptimal classes. An initial model was generated from the resulting 351,941 particles, and a 3D classification with image alignment was performed to find the most populated state. Particles contributing to this state underwent subsequent consensus refinement without symmetry, followed by post-processing, which yielded an 8.3 Å volume. To obtain an improved single-channel volume, focused refinement was performed on the two best-looking channels, leading to a 7.7 Å volume more clearly indicating a homo-heptameric assembly per MscS2 channel. (TIF)

**S2 Fig. Cryo-EM data processing attempts that did not improve the MscS2 superstructure or single-channel volume.** Initially, 2000 micrographs were collected and manually inspected in RELION 3.1 and 4.0 (black branch) to remove any affected by drift, poor CTF fits, or low-

quality ice, resulting in a total of 1856 micrographs. Particles were picked through Laplacian auto-picking and underwent one round of 2D classification to eliminate picking contaminants. An initial model was generated from the resulting 300,757 particles. A subsequent 3D classification with D3 symmetry, followed by a consensus refinement of particles in the best class, led to an 8 Å volume lacking connectivity and proper features. Consensus refinement of the 300,757 particles without symmetry, followed by post-processing, yielded an 8.3 Å volume. Focused refinement of a single heptamer led to a featureless 7 Å volume. 3D classification of three heptamers in the C1 refined volume gave no additional information. Later, RELION 5.0 beta (green branch) was used to generate 2D class averages for reference-based picking from all 1856 micrographs. The extracted particles were subjected to multiple rounds of 2D classification to sort out suboptimal classes, followed by 3D classification. Particles of the most populated class 4 underwent consensus refinement with D3 symmetry, resulting in a 7.5 Å volume. However, the features are not as pronounced as in our published volume. Next, the 1856 motion-corrected micrographs were imported into CryoSPARC (blue branch), and CTF estimation and manual exposure correction were performed. From the resulting 1793 micrographs, 393 455 particles were picked and subjected to 2D classification for 2D-class average-based template generation. Template picking from the 1793 micrographs yielded 1,279,465 particles. Picking contaminants and suboptimal classes were filtered through consecutive 2D classification, resulting in 120,028 particles subjected to both C1 and D3 symmetry-based non-uniform refinement, respectively. The obtained volumes had incomplete features and showed insufficient connectivity. Further, all C1 refinements indicate that the superstructure lacks inherent symmetry, and despite multiple attempts to impose D3 symmetry during processing, the obtained volumes fail to show symmetric features. None of the methods improved the resolution or quality of the experimental volume or sub-volume.
(TIF)

**S3 Fig. Multiple sequence alignment of *E. coli* MscS and 35 microsporidian MscS2, *N. displodere* MscS2 AlphaFold2-model and hydropathicity prediction. A)** Multiple sequence alignment (MSA) of *E. coli* MscS and microsporidian MscS2 shows high conservation of the MscS β-domain and moderate conservation of the transmembrane helix 3b (TMH3b). MSA was generated using ClustalW [42] with default settings, colored according to conservation with "Clustal" in Jalview (v2.11.2.6) and set to black and white. **B)** AlphaFold2 (v2.3.1) prediction of homo-heptameric *N. displodere* MscS2 (left), colored by pLDDT confidence measure, with indicated color key, and predicted aligned error plots (right). **C)** Hydropathicity prediction for *N. displodere* MscS2 to identify hydrophobic regions, calculated using ProtScale on the ExPASy server [49].
(TIF)

**S4 Fig. Test production of full-length MscS2 and purification of truncated MscS2. A)** Western Blot analysis of induced and non-induced cultures comparing MscS2 yields from *A. locustae*, *E. bieneusi*, *E. cuniculi*, and *N. displodere* produced with a C-terminal His10 tag in *E. coli* Rosetta (DE3). Anti-His6x antibody from mouse was used for immunoprecipitation (1:1000). GFP-MS2-His10 construct served as positive control for protein production and antibody binding. As seen in the blots, the promotor is leaky as MscS2 is produced to some extent in the absence of IPTG. **B)** IMAC purification of DDM-solubilized His14-SUMO-MscS2Δ2–29 visualized by SDS-PAGE showing that the protein can be produced in high amounts. **C)** SDS-PAGE analysis of the fractions corresponding to the reverse IMAC of DDM-solubilized MscS2Δ2–29 post tag cleavage. Fraction used for negative-stain TEM studies is boxed in black and fractions analyzed via SEC are marked with blue boxes. kDa, kilo Dalton; M, marker; IPTG, Isopropyl β-d-1-thiogalactopyranoside; GFP, green-fluorescent protein; MS2, MS2

bacteriophage coat protein; pooled E, pooled elution; FT, flowthrough; SEC, size exclusion chromatography, kDa, kilo Dalton; M, marker; FT, flowthrough; W, wash; E, elution; SUMO, small ubiquitin-related modifier; SEC, size-exclusion chromatography.
(TIF)

**S5 Fig. Tag cleavage of truncated MscS2 induces an increase in molecular weight, indicating higher oligomeric complex formation. A**) Size exclusion chromatogram (SEC) profile of DDM and LMNG-solubilized MscS2Δ2–29 with and without (only for DDM) His14-SUMO tag. DDM-solubilized MscS2Δ2–29 elutes at 10.16 mL, His14-SUMO-MscS2Δ2–29 in DDM elutes at 10.87, and the latter solubilized with LMNG elutes at 10.93 mL. The observed main peak shift post tag cleavage is indicated. The x-axis of the SEC profile shows the elution volume in milliliters (mL), the y-axis displays the absorption in milli-absorbance units, and a second x-axis at the top shows the molecular weight of standard globular proteins. **B-D**) SDS-PAGE analyses of the indicated SEC-elution fractions for B) MscS2Δ2–29 (ca. 9 kDa) and C) His14-SUMO-MscS2Δ2–29 (ca. 23 kDa), both solubilized using DDM and D) His14-SUMO-MscS2Δ2–29 solubilized with LMNG. The main SEC peaks in A) correspond to MscS2Δ2–29 with or without His14-SUMO tag. **E**) Mass photometry analysis comparing the molecular weight of MscS2Δ2–29 with and without His14-SUMO tag. For His14-SUMO-MscS2Δ2–29 (12nM), a main peak corresponding to 158 (± 15.6) kDa was detected, and MscS2Δ2–29 (140 nM) produced a main peak at 452 (± 112) kDa. **F**) Buffer control of the mass photometry experiments showing a peak around 30 (± 4.8) kDa and 61 (± 15.6) kDa.
(TIF)

**S6 Fig. *mscS2* transcription in different life-cycle stages of *Vavraia culicis*, *Edhazardia aedis*, *Nematocida parisii* and *Encephalitozoon cuniculi*. A,B**) Gene expression pattern of *mscS2* in two microsporidians, *Vavraia culicis* (A, VCUG_00285) with a simple, and *Edhazardia aedis* (A, EDEG_01910) with a complex life cycle [69]. The life-cycle stages corresponding to the sampling points are indicated. Each developmental stage was sampled in duplicates [69] and is shown in shades of yellow (A) and cyan (B). **C**) Transcript levels of *Nematocida parisii* *mscS2* (NEPG_00116) over the course of a full infection cycle: 1) 8 hours post infection (hpi) sporoplasm stage, 2) 16 hpi early meront stage, 3) 30 hpi late meront stage, 4) 40 hpi onset of spore formation and 5) 64 hpi spores within membrane-bound vesicles. Samples from 8, 16, and 30 hpi are reported to be dominated by proliferating meronts, and later samples at 40 and 64 hpi harbor a mixture of meront, sporont and mature spore stages. Developmental stages were assessed by differential interference contrast microscopy and fluorescence in situ hybridization [1]. **D**) Transcriptional profile of mscS2 in *Encephalitozoon cuniculi* (Ecu09_0470) after 24, 48, and 72 hpi. Transcriptomic data indicates that at 24 hpi, proliferation rates are throttled due to the downregulation of housekeeping genes. Further, at 48 hpi, meronts start producing spore-related genes, but spore formation is not expected until after 72 hpi [70].
(TIF)

**S1 Table. Primers used in this study.**
(TIF)

**S2 Table. Cryo-EM data collection and processing.**
(TIF)

# Acknowledgments

The authors thank Prof. Robert Hirt and Dr. Josy ter Beek for interesting discussions regarding the project. We thank Dr. Michael Hall and Camilla Holmlund for help with cryo-EM data

collection and Dennis Svedberg for aid with the cladogram. Further, we thank the previous Barandun Lab for constructive discussions. The electron microscopy data was collected at the Umeå Core Facility for electron Microscopy, a node of the Cryo-EM Swedish National Facility, funded by the Knut and Alice Wallenberg, Family Erling Persson and Kempe Foundations, SciLifeLab, Stockholm University, and Umeå University.

## Author Contributions

**Conceptualization:** Alexandra Berg, Jonas Barandun.

**Funding acquisition:** Ronnie P.-A. Berntsson, Jonas Barandun.

**Investigation:** Alexandra Berg.

**Project administration:** Alexandra Berg.

**Resources:** Ronnie P.-A. Berntsson, Jonas Barandun.

**Supervision:** Ronnie P.-A. Berntsson, Jonas Barandun.

**Validation:** Alexandra Berg.

**Visualization:** Alexandra Berg.

**Writing – original draft:** Alexandra Berg.

**Writing – review & editing:** Alexandra Berg, Ronnie P.-A. Berntsson, Jonas Barandun.

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
