## [Decision Letter · Decision Letter 0]

9 May 2024

PONE-D-24-12059Nematocida displodere Mechanosensitive Ion Channel of Small Conductance 2 assembles into a unique 6-channel super-structure in vitroPLOS ONE

Dear Dr. Berg,

Thank you for submitting your manuscript to PLOS ONE. After careful consideration, we feel that it has merit but does not fully meet PLOS ONE’s publication criteria as it currently stands. Therefore, we invite you to submit a revised version of the manuscript that addresses the points raised during the review process. Please submit your revised manuscript by Jun 23 2024 11:59PM. If you will need more time than this to complete your revisions, please reply to this message or contact the journal office at plosone@plos.org. In your revised manuscript you will need to address as fully as possible the detailed and cogent criticisms raised by Reviewer 1.

Please include the following items when submitting your revised manuscript:A rebuttal letter that responds to each point raised by the academic editor and reviewer(s). You should upload this letter as a separate file labeled 'Response to Reviewers'.A marked-up copy of your manuscript that highlights changes made to the original version. You should upload this as a separate file labeled 'Revised Manuscript with Track Changes'.An unmarked version of your revised paper without tracked changes. You should upload this as a separate file labeled 'Manuscript'.

We look forward to receiving your revised manuscript.

Kind regards,

Israel Silman

Academic Editor

PLOS ONE

Journal Requirements:

Reviewers' comments:

Reviewer's Responses to Questions

**Comments to the Author**

1. Is the manuscript technically sound, and do the data support the conclusions?

Reviewer #1: No

Reviewer #2: Yes

2. Has the statistical analysis been performed appropriately and rigorously? 

Reviewer #1: N/A

Reviewer #2: N/A

3. Have the authors made all data underlying the findings in their manuscript fully available?

Reviewer #1: Yes

Reviewer #2: Yes

4. Is the manuscript presented in an intelligible fashion and written in standard English?

Reviewer #1: Yes

Reviewer #2: Yes

5. Review Comments to the Author

Reviewer #1: In this manuscript, Berg et al. study mechanosensitive ion channel of small conductance 2 (MscS2) from the microsporidium Nematocida displodere. Sequence analysis and AlphaFold modeling suggest that MscS2 lost most of the cytoplasmic domain of the archetypal MscS channel, retaining only the beta-domain, as well as most of the transmembrane domain, retaining possibly only a TM3-like helix, which forms the ion-conducting channel in MscS. After construct optimization, the authors purify an N-terminally truncated MscS2 from Nematocida displodere. Cryo-EM analysis yielded a density map at 8.3 Å resolution and showed that six MscS2 channels assemble into a superstructure resembling a six-way cross-joint. The authors interpret density for one of these channels using their AlphaFold model and conclude that MscS2 forms the expected heptamer. The authors further draw conclusions from the comparison of their 8.3 Å map with the structure of MscS, concluding that MscS2 is compacted compared to MscS. However, the authors recognize that - if MscS2 would indeed function as a mechanosensitive channel - the tension-sensing mechanism would have to be very different from that of MscS.

The study is technically done okay. However, the image processing that was performed is very basic and no advanced techniques, such as symmetry expansion, 3D variance analysis, masking of just one channel or substructure etc., were applied in order to obtain a higher-resolution map of the channel. I would expect more effort in image processing to be expended before publishing this rather low-resolution structure.

The paper could also be presented much better. The introduction describes microsporidia in great detail, but this information is only needed for one supplementary figure (Fig. S5). On the other hand, the introduction lacks much of the known information on the structure and gating of MscS and, more importantly, does not discuss the many structural variations found in MscS-like channels. In fact, it is not at all clear whether MscS2 functions as a mechanosensitive channel, and it is structurally so different from MscS, especially the transmembrane domain, that it may be better to refer to MscS2 as an MscS-like channel rather than as an MscS channel. Basically, since this is a structural paper, the reader would be much better served if the introduction would provide more information on the structure of MscS and structural features of MscS-like channels and less information on microsporidia.

There are several points the authors have to address before this manuscript would be suitable for publication:

The authors conclude from their sequence and AlphaFold analyses that “with TMH3 and the conserved beta-domain, the microsporidian MscS2 maintains the essential domains of a functional mechanosensitive ion channel”. This conclusion is not justified. Most people consider helices TM1 and TM2 to be the tension-sensing module. Without these helices, it is not at all obvious that MscS2 could function as a mechanosensitive ion channel. To be able to state that MscS2 functions as a mechanosensitive (and not some other type of) ion channel, the authors would have to perform electrophysiology.

Figure 4 seems to be extremely misleading. It is very unusual to use such similar colors for the map and the model. It would be much more informative if the authors would use a high-contrast color for the model and a low-contrast color for the transparent surface. The authors also do not point out that the cryo-EM density (as an aside cryo-EM does not provide electron density - that would be X-ray diffraction) mostly just accounts for the cytoplasmic domain of MscS2 but not for any of the transmembrane helices. Another complication the authors do not address is that TM3a in MscS, despite its name, is actually not a transmembrane helix. It is in fact outside of the membrane, and it is in fact the ends of helices TM1 and TM2 that incorporate the channel into the membrane (the authors could indicate the membrane-spanning region for MscS and they will immediately notice the problem). Thus, the EcMscS 92-182 model shown in Fig. 4B would actually not have a transmembrane domain (if TM3a would be the same in MscS2 as in EcMscS). It seems very difficult to imagine that TM3a, which only forms the ion-conduction pathway (outside the membrane) in EcMscS, would serve the same purpose in MscS2 as well as have to incorporate into the membrane and also have sense membrane tension.

The authors discuss that MscS2 is greatly compacted compared to archetypal MscS and misses the tension-sensing helices, and so they conclude that MscS2 may sense tension differently from MscS2. However, given the substantially different transmembrane domain and since there are no functional data, it is not that far-fetched to assume that the function of MscS2 may actually have evolved away from being a mechanosensitive channel and may in fact NOT be a mechanosensitive channel at all. At this point, the function of MscS2 is not defined.

The mscS2 gene is predominantly upregulated during proliferative stages of microsporidia and is only moderately expressed during stages that require morphological changes (Fig S5). This seems to go against the idea that MscS2 functions mainly as a mechanosensitive channel regulating homeostasis, as the expected expression pattern would then be more like the opposite.

Minor points:

In the Abstract and Introduction, the authors write that microsporidia have at least 5 different mscS copies. However, the words “copies” and “different” are mutually exclusive because copies are, by definition, identical.

Figure 2B: It should be 28/56 “nM” not 28/56 “nm”.

Line 314: “MscS particles from E. coli visualized via TEM (56)”. The title of reference 56 is “Structural mechanism for gating of a eukaryotic mechanosensitive channel of small conductance”. E. coli is not a eukaryote, so this reference makes no sense.

Reviewer #2: Berg at al. describe an unusual oligomeric structure of a small variant of a mechanosensitive channel of small conductance. This channel lacks key elements of the prototypic bacterial MscS, such as a complete transmembrane sensor paddle and an enclosed cytosolic vestibule. The authors have used electron cryo microscopy for structure determination at intermediate resolution (sub-nanometer). The structure shows that 6 heptameric channels assembled into a cube-like super structure. This finding is unexpected and curious but probably without biological relevance. The authors relate the super structure to the C-terminal truncation of the rudimentary sensor paddle, which they introduced to improve expression and purification.

The manuscript is well written. It combines solid biochemical studies for establishing the oligomerization state, with electron microscopy for structure determination. The findings are complemented by sequence analysis for establishing the relation to other mechanosensitive channels.

To my knowledge, the authors present the 1st structure of a MscS-like channel with a reduced vestibule with an accessible pore. However, the modest resolution of the map is a missed opportunity for gaining more in depth understanding how ion selectivity is achieved in such a channel.

6. PLOS authors have the option to publish the peer review history of their article (what does this mean?). If published, this will include your full peer review and any attached files.

Reviewer #1: No

Reviewer #2: No

---

## [Author Response · Author response to Decision Letter 0]

2 Jul 2024

Please see our uploaded "Response to reviewers" where we have addressed all comments and thank the reviewers for their time and feedback.

---

## [Decision Letter · Decision Letter 1]

10 Jul 2024

Nematocida displodere Mechanosensitive Ion Channel of Small Conductance 2 assembles into a unique 6-channel super-structure in vitro

PONE-D-24-12059R1

Dear Dr. Berg,

We’re pleased to inform you that your manuscript has been judged scientifically suitable for publication and will be formally accepted for publication once it meets all outstanding technical requirements.

Kind regards,

Israel Silman

Academic Editor

PLOS ONE

Additional Editor Comments (optional):

Reviewers' comments:

Reviewer's Responses to Questions

**Comments to the Author**

1. If the authors have adequately addressed your comments raised in a previous round of review and you feel that this manuscript is now acceptable for publication, you may indicate that here to bypass the “Comments to the Author” section, enter your conflict of interest statement in the “Confidential to Editor” section, and submit your "Accept" recommendation.

Reviewer #1: All comments have been addressed

2. Is the manuscript technically sound, and do the data support the conclusions?

Reviewer #1: Yes

3. Has the statistical analysis been performed appropriately and rigorously? 

Reviewer #1: N/A

4. Have the authors made all data underlying the findings in their manuscript fully available?

Reviewer #1: Yes

5. Is the manuscript presented in an intelligible fashion and written in standard English?

Reviewer #1: Yes

6. Review Comments to the Author

Reviewer #1: The authors have addressed all my concerns. One last comment is that it is hard to believe that symmetry expansion, when done correctly, would not have substantially improved the resolution of the map, especially when applying 7-fold symmetry to the map obtained by symmetry expansion. However, even with these tricks, it is unlikely that the resulting map would have provided more insight into the architecture of the transmembrane domain. Therefore, I support publication of this manuscript in the current form.

7. PLOS authors have the option to publish the peer review history of their article (what does this mean?). If published, this will include your full peer review and any attached files.

Reviewer #1: No

---

## [Editor Report · Acceptance letter]

12 Jul 2024

PONE-D-24-12059R1 

PLOS ONE

Dear Dr. Berg, 

I'm pleased to inform you that your manuscript has been deemed suitable for publication in PLOS ONE. Congratulations! Your manuscript is now being handed over to our production team.

Kind regards, 

on behalf of

Prof. Israel Silman 

Academic Editor

PLOS ONE